# Optimizing Deep CNN Architectures for Face Liveness Detection

**DOI:** 10.3390/e21040423

**Published:** 2019-04-20

**Authors:** Ranjana Koshy, Ausif Mahmood

**Affiliations:** Computer Science and Engineering Department, University of Bridgeport, Bridgeport, CT 06604, USA

**Keywords:** face liveness detection, nonlinear diffusion, NUAA dataset, CNN-5, ResNet50, Inception v4

## Abstract

Face recognition is a popular and efficient form of biometric authentication used in many software applications. One drawback of this technique is that it is prone to face spoofing attacks, where an impostor can gain access to the system by presenting a photograph of a valid user to the sensor. Thus, face liveness detection is a necessary step before granting authentication to the user. In this paper, we have developed deep architectures for face liveness detection that use a combination of texture analysis and a convolutional neural network (CNN) to classify the captured image as real or fake. Our development greatly improved upon a recent approach that applies nonlinear diffusion based on an additive operator splitting scheme and a tridiagonal matrix block-solver algorithm to the image, which enhances the edges and surface texture in the real image. We then fed the diffused image to a deep CNN to identify the complex and deep features for classification. We obtained 100% accuracy on the NUAA Photograph Impostor dataset for face liveness detection using one of our enhanced architectures. Further, we gained insight into the enhancement of the face liveness detection architecture by evaluating three different deep architectures, which included deep CNN, residual network, and the inception network version 4. We evaluated the performance of each of these architectures on the NUAA dataset and present here the experimental results showing under what conditions an architecture would be better suited for face liveness detection. While the residual network gave us competitive results, the inception network version 4 produced the optimal accuracy of 100% in liveness detection (with nonlinear anisotropic diffused images with a smoothness parameter of 15). Our approach outperformed all current state-of-the-art methods.

## 1. Introduction

Biometric authentication is a well-known security process used to ensure secure access to digital computing devices. The authentication system determines the individual’s identity based on biological characteristics that are unique to the individual. Some of the popular authentication schemes include fingerprint scan, retina scan, iris recognition, speaker recognition, hand and finger geometry, vein geometry, voice identification, and so forth.

Face recognition is a popular biometric authentication technique used for identity management and secure access control for many web- and mobile-related software applications. It is more convenient to deploy than other biometric techniques. However, despite its advantage as a nonintrusive form of access, the security system might not be able to distinguish between a real person and his or her photograph. An impostor can gain access to the system by presenting a copy of the image to the camera. Therefore, prior to face recognition authentication, face liveness detection is important to detect whether the captured face is live or fake. To address face spoofing attacks, researchers have proposed different methods for face liveness detection, for example, based on enhanced local binary pattern (LBP), motion analysis, texture analysis, quality of captured images, and so forth. Recent research has focused on using deep convolutional neural network (CNN) architectures for face liveness detection.

In this paper, we present different CNN architectures that perform face liveness detection on images captured through a webcam by first diffusing the images and then feeding them to the CNN architectures. We evaluated the performance of each of these architectures on the NUAA Photograph Impostor dataset to determine an optimal architecture for liveness detection. We determined that the level of smoothness of the diffused images played a significant role in determining the liveness of the captured images. We experimented with various values of the parameter param. alpha, which defines the smoothness of diffusion, and observed that a reasonably low value of 15 or 25 enhanced the edges and boundaries of the image and gave better results, whereas higher values blurred out important information from the image. With a param. alpha value of 15, the inception network version 4 (Inception v4) outperformed the residual network and a simple five-layer CNN (CNN-5). With a param. alpha value of 25, the CNN-5 and residual network performed slightly better than the inception network. We also performed a comparison of our proposed approach with previous state-of-the-art approaches, as well as other deep architectures for face liveness detection, and demonstrated that the Inception v4, when applied to processed images with nonlinear diffusion, results in optimal performance in the detection of image liveness to counteract face spoofing.

The rest of the paper is organized as follows: Section 2 discusses previous work related to face liveness detection. Our proposed method of using CNN architectures on nonlinear diffused images is discussed in Section 3. Section 4 presents a performance evaluation of the different CNN architectures we used for face antispoofing. The concluding remarks are mentioned in Section 5.

## 2. Related Work 

Many methods have been used to address face spoofing attacks to determine the liveness of a captured image. Liu et al. [1] presented the extraction of an enhanced local binary pattern of a face map that served as the classification features. These features, when fed to a Support Vector Machine (SVM) classifier, can identify whether the face map is real or fake. Das et al. [2] described an approach based on frequency and texture analyses for differentiating between live and fake faces. The frequency analysis is performed by transforming the images into the frequency domain using Fourier transform and then calculating the frequency descriptor to determine the temporal changes in the face. The texture analysis is done using LBP, and the resulting feature vector is fed to an SVM classifier with a radial basis function kernel for classification. In the proposed method by Kim et al. [3], the difference in surface properties between live and fake faces is estimated by computing the diffusion speed and extracting antispoofing features based on the local patterns of diffusion speeds. These features are fed to a linear SVM classifier to determine the liveness of the image.

Yeh et al. [4] proposed an algorithm that relies on the property of digital focus with various depths of field. Preprocessing is done by analyzing the nose and bottom right part of the cheek. The level of blurriness is shown to be different in live and fake images due to the effect of the depth of field. Classification is then done using the k-nearest-neighbor algorithm. A method that uses a flash against 2D spoofing attack was described by Chan et al. [5]. In this method, two images per person are captured—one using flash and the other without using flash. The textural information from the face is measured using a descriptor based on uniform LBP, and three other descriptors are used to capture the structural information of the face using the standard deviation and mean of the grayscale difference between the two captured images of a person. Classification is based on the difference between the images with and without flash measured by the four descriptors. Luan et al. [6] proposed a method where three types of features are extracted, which are specular reflection ratio, hue channel distribution features, and blurriness. Classification of the images is done based on these three features using an SVM. Tan et al. [7] presented a method where the antispoofing task was formulated as a binary classification problem. They analyzed the 2D Fourier spectra by Difference of Gaussian (DoG) filtering to remove noise and extended the sparse logistic regression classifier nonlinearly and spatially for classification. The authors of [8] introduced a texture descriptor known as the dynamic local ternary pattern, where the textural properties of facial skin are explored using a dynamic threshold setting, and the support vector machine with a linear kernel is used for classification. Matta et al. [9] analyzed the texture of facial images using multiscale local binary patterns. The microtexture patterns were then encoded into an enhanced feature histogram, which was fed to an SVM classifier.

Recent work in face liveness detection has been based on the use of deep CNN architectures [10,11] as these provide better liveness detection accuracy than the previously mentioned approaches. The work proposed in [10] focused on training deep CNNs for liveness detection by employing data randomization on small minibatches. The research proposed in [11] utilized a combination of diffusion of the input face followed by only a three-layer CNN architecture. None of the existing works has employed deeper architectures such as the ResNet and the Inception v4 that we employed in this paper. 

## 3. Proposed Method

In our method, we proposed a solution where we first applied nonlinear diffusion based on an additive operator splitting scheme and an efficient block-solver called the tridiagonal matrix algorithm to the captured image in order to enhance the edges and preserve the boundary locations of the image (similar to the work proposed in [11] but also with deeper architectures). These diffused input images were then fed to the CNN architecture to extract the complex and deep features and to finally classify the image as real or fake. We used three different CNN architectures and performed a comparative evaluation of their performance, thus gaining insight into why a particular architecture is better suited for face liveness detection. Additionally, we compared our proposed method with previous approaches that have been proposed for liveness detection to determine how well it performed with regard to current state-of-the-art methods.

### 3.1. Nonlinear Diffusion

Nonlinear diffusion is a powerful denoising technique which denoises an image by preserving the edges. The denoised image is a solution of the diffusion equation with a diffusion coefficient that varies spatially. The information contained in high-spatial-frequency components of the image is preserved by the diffusion process [12]. Nonlinear diffusion applied to face liveness detection helps in distinguishing a fake from a real image by diffusing the input image quickly. The edges obtained from a fake image will be faded, while those obtained from a real image will remain clear [13]. Nonlinear diffusion based on a partial differential equation, called anisotropic diffusion, helps prevent the problem of blurring important features and localization issues associated with linear diffusion. According to Perona and Malik [14], anisotropic diffusion is used to improve the scale-space technique, where the diffusion coefficient is chosen to vary spatially such that inter-region smoothing is encouraged in preference to intraregion smoothing, thereby enhancing the boundaries.

During the diffusion process, the nonlinear diffusion filter detects the edges and preserves the locations using explicit schemes. A semi-implicit scheme proposed in [15] addressed the problem of regularization associated with anisotropic diffusion. This is the additive operator splitting (AOS) scheme, which is stable for all time steps and guarantees equal treatment of all coordinate axes, defined by Equation (1) [16]. This scheme enables fast diffusion, smooths out the edges in fake images, and preserves them in real ones. Image information inside objects will be blended, while image information along edges will be left intact.
(1)$${\left(I_{k}\right)}^{t+1}=\overset{m}{\underset{l=1}{\sum }}{\left(mId-\tau m^{2}A_{l}\right)}^{-1}\;I_{k}^{t}$$
where *m* is the number of dimensions, *k* represents the channel, *Id* is the identity matrix, $A_{l}$ is the diffusion, and τ is the time steps (referred to as param. alpha in our implementation). For a two-dimensional case, m = 2, and the equation becomes
(2)$${\left(I_{k}\right)}^{t+1}={\left(2Id-4\tau A_{1}\right)}^{-1}I_{k}^{t}+{\left(2Id-4\tau A_{2}\right)}^{-1}I_{k}^{t}$$
where $A_{1}$ and $A_{2}$ denote the diffusion in the horizontal and vertical directions.

AOS, together with a block-solver tridiagonal matrix algorithm (TDMA), can be used to efficiently solve the nonlinear, scalar valued diffusion equation [16]. The TDMA is a simplified form of Gaussian elimination, useful for solving tridiagonal systems of equations.

### 3.2. CNN Architectures

CNNs have proved successful in many machine learning tasks such as handwriting recognition [17], natural language processing [18], text classification [19], image classification [17], face recognition [20], face detection [21], object detection [22], video classification [23], object tracking [24], super resolution [25], human pose estimation [26], and so forth. CNNs, introduced by Lecun et al. [17], combine three architectural concepts of local receptive fields, shared weights, and spatial or temporal subsampling in order to ensure some degree of shift, scale, and distortion invariance. They eliminate the need for hand-crafted feature extraction and extract local features by restricting the receptive fields of hidden units to be local.

The CNN architecture takes an image as input, performs the required number of convolutions and subsampling, and then feeds the outputs obtained to the required number of fully connected layers.

The CNN architectures used in our experiments are described below.

#### 3.2.1. CNN-5

This consisted of a total of five layers, which included two convolutional layers, one subsampling layer, and two fully connected layers. The inputs to the network were the 64 × 64 size nonlinear diffused counterparts of the captured original images. The first layer was the convolutional layer C1 that consisted of 12 feature maps, each of size 56 × 56, where each unit in a feature map was the result of the convolution of the local receptive field of the input image with a 9 × 9 kernel. The kernel used in the convolution was the set of connection weights used by the units in the feature map. Each unit in a feature map shared the same set of weights and bias, so they detected the same feature at all possible locations in the input. Other feature maps in the layer used different sets of weights and biases, extracting different local features [17]. The next layer was also a convolutional layer, C2, consisting of 18 feature maps, each of size 50 × 50, obtained by the convolution of the feature maps in C1 with 7 × 7 kernels. C2 was followed by a subsampling layer S2 of 18 feature maps, each of size 25 × 25, obtained by applying a pooling of size (2, 2) to the corresponding C2 layer feature maps, thereby reducing the resolution of the feature maps in the C2 layer by half. The next layer was a fully connected hidden layer of 50 neurons, followed by a fully connected output layer of 2 neurons. Dropout of probability 0.25 was applied to the pooling layer, and dropout of 0.4 was applied to the hidden layer. The complete architecture is illustrated in Figure 1.

To introduce nonlinearity into the model, the rectified linear unit activation function was applied to the outputs of C1, C2, and the hidden layer of 50 neurons, restricting the outputs of these layers to be 0 or x, where x is the output of the neuron before the activation function is applied. The sigmoid activation function was applied to the output layer, giving an output in the range of 0 to 1. The network was trained by backpropagation using the Adam optimization algorithm, with mean squared error as the loss function. The diffusion block was implemented via direct implementation of the diffusion equations.

#### 3.2.2. ResNet50

As the depth of a deep network increases, the accuracy becomes saturated and the network degrades rapidly. The deep residual learning framework improves the degradation problem and eases the training of the network [27]. In a residual network, there are shortcut connections which skip one or more layers. To the standard CNN, skip connections are added that bypass a few convolution layers. Each bypass produces a residual block where the convolution layers predict a residual that is added to the block’s input. Therefore, these shortcut connections simply perform identity mapping, and their outputs are added to the outputs of the stacked layers. This gives rise to a shallower architecture, and the entire network can still be trained end-to-end. The identity connections add neither extra parameters nor computational complexity to the network. Residual networks have been used successfully in age and gender estimation [28], for hyperspectral image classification [29], and other classification tasks. Even though very deep residual networks (152 layers) have been used, in this work, we only used 50 layers, as our focus was on a comparative evaluation of architectures.

The ResNet50 is a residual network of 50 layers, which consists of identity residual blocks and convolutional residual blocks stacked together. The shortcut connections allow the gradient to be back-propagated to earlier layers, preventing the vanishing gradient problem associated with very deep networks. The central idea of residual networks (ResNets) is to learn the additive residual function, using an identity mapping realized through identity skip connections [30]. The shortcut connections parallel to the main path of convolutional layers allow the gradient to back-propagate through them, resulting in faster training.

Figure 2 shows the basic residual block, and Figure 3 shows the ResNet50 architecture used in this paper.

The architecture consisted of identity blocks and convolutional blocks stacked together in stages 2–5. The identity residual block was used when the input and output dimensions were same before the addition, whereas the convolutional residual block was used when the input and output dimensions were different before the addition. 

The inputs to the network were the 64 × 64 nonlinear diffused images. The network was trained using the Adam optimization algorithm, with categorical cross-entropy as the loss function and softmax as the classifier. Cross-entropy measures the performance of a classification model where the output is a probability distribution, giving a value between 0 and 1. Since our targets were in categorical format with two classes (fake, real), we used the categorical_crossentropy as the loss function.

#### 3.2.3. Inception v4

Inception architectures have been shown to achieve very good performance at relatively low computational cost [31]. The inception network’s architecture provides improved utilization of the computing resources inside the network through a design that increases the depth and width of the network while keeping the computational budget constant. The network consists of inception modules stacked upon each other, with occasional pooling layers to reduce the resolution of the grid. Information is processed at various scales by applying filters of different sizes to the same layer, which are then aggregated so that the following stage can abstract features from different scales simultaneously [32]. 

The basic inception module (Figure 4) used filter sizes 1 × 1, 3 × 3, and 5 × 5, the convolution outputs and a max pooling output of which were concatenated into a single output vector forming the input to the next stage. Instead of using a filter of an appropriate size at a level, filters of multiple sizes were used, making the network wider than deeper, so that features at different scales could be recognized. The resulting feature maps were then concatenated before going to the next layer. This was further refined by incorporating dimensionality reductions by applying 1 × 1 convolutions before the 3 × 3 and 5 × 5 convolutions. Auxiliary classifiers were also included to prevent the vanishing gradient problem. Later versions incorporated factorization of filters into smaller convolutions by replacing the 5 × 5 filters with two 3 × 3 filters [33]. This reduced the computational cost, resulting in a reduced number of parameters and faster training. Additionally, the nxn convolutions were replaced by a 1xn convolution followed by a nx1 convolution, which further increased the computational cost savings. Also, filter banks were expanded to avoid representational bottleneck. Batch normalization was applied to the fully connected layer of auxiliary classifiers for additional regularization, and label smoothing was added as regularization to the loss formula to make the model more adaptable and prevent overfitting. Deep CNN with inception modules has been used for person reidentification [34], inception network version 3 has been used for Next Generation Sequencing (NGS)-based pathogen detection [35], and a residual inception network has been used in multimedia classification [36]. Figure 4 below shows the basic inception module.

In our experiments, we used the Inception v4 architecture, which is a more uniform simplified structure compared with the earlier versions. This model consisted of three different inception blocks, where each was used repeatedly a certain number of times, and two reduction blocks for changing the width and height of the grid. The inputs to the network were the 64 × 64 nonlinear diffused images. The complete network consisted of an inception stem, 4 x inception A blocks, 1 x reduction A block, 7 x inception B blocks, 1 x reduction B block, and 3 x inception C blocks, followed by average pooling, dropout, and softmax layers. Figure 5 shows the Inception v4 network.

## 4. Performance Evaluation

We performed experimental evaluations using the three CNN architectures (CNN-5, ResNet50, and Inception v4) on the NUAA dataset. In this section, we describe the experimental results and the hyperparameter settings used with each of the architectures in the experiments. We compare the three architectures in terms of performance and, further, compare their performance with other existing liveness detection methods.

### 4.1. Dataset

The NUAA Photograph Impostor database [7] contains images of 15 subjects and is publicly available. It contains both live and photographed faces of the subjects, captured using a webcam in three sessions, with each session at a different place with different illumination conditions. There is a total of 12,614 images in the dataset, with 3491 training images and 9123 test images, as shown in Table 1. Each image is 64 × 64 in size and in gray-scaled representation. There is no overlap between the training set and test set images. The training set images were taken during sessions 1 and 2, and the test images were taken during session 3. The training set contain images of subjects 1–9, whereas the test set contains images of subjects 1–15.

### 4.2. Experimental Setup

We created nonlinear diffused images using the MATLAB implementation code by the author of [16]. Most of our experiments were implemented on a 32GB RAM PC. The code was written in Python, using the Keras library with the TensorFlow backend, for all the three architectures.

### 4.3. Experimental Results

We conducted numerous experiments with the three CNN architectures on the NUAA dataset. We performed several tests by changing the hyperparameters during the learning phase. 

From all the experiments we conducted, we figured out that the smoothness of the nonlinear diffused images played a significant role in determining the liveness of the image. A high value of 50/75/100 for the smoothness parameter param. alpha blurred out important edges from the images, as depicted in Figure 6. A value of 15 or 25 produced better diffused images with highlighted edges and enhanced boundaries.

#### 4.3.1. CNN-5

With the CNN-5 architecture, we used diffused images created by setting the parameter that defines smoothness of diffusion (param. alpha) to 25. We obtained a test accuracy of 95.99% after training for 30 epochs using the Adam optimizer and mean-squared-error loss function. The learning rate was set to 0.001, which is the default of the Adam optimizer, and the batch size was set to 32. The activation functions used were ReLU for the convolutional layers and hidden layer, and sigmoid for the output layer. Average pooling was applied to the C2 layer for down-sampling. Table 2 shows the test accuracies obtained with CNN-5 after training for various epochs.

A slightly lower accuracy of 94.78% was obtained when max pooling was used after the C2 layer, categorical cross-entropy as the loss function, and softmax as the classifier. This accuracy was obtained after training for 35 epochs, with Adadelta as the optimizer at its default learning rate and with the batch size set to 32.

#### 4.3.2. ResNet50

With the ResNet50 architecture, various combinations of kernel initialization, loss functions, activation functions, and classifiers were tested. We used diffused images created by setting the smoothness of diffusion (param. alpha) to 25.

We obtained a test accuracy of 95.85% after training for 20 epochs using the Adam optimizer, categorical cross-entropy loss function, and softmax classifier. The learning rate was kept at the default of the Adam optimizer, which is 0.001, and the batch size was set to 32. Table 3 shows the accuracies obtained with ResNet50 for various epochs.

We also tested for liveness detection by reducing the number of feature maps in the convolutions by a quarter. We tested the architecture with Adam, Adadelta, RMSprop, and SGD optimizers, and the best results were obtained with the Adam optimizer. We also experimented with two different kernel initializers: glorot-uniform() and random-uniform(). It was observed that, on average, the glorot-uniform() initializer gave better results than the random-uniform() initializer. We further observed that the network with the original number of feature maps in the convolutions gave better results than the network with a reduced number of feature maps.

#### 4.3.3. Inception v4

With the Inception v4 architecture, we achieved the highest test accuracy of 100%, something which has not been reported by any previous approach for face liveness detection on the NUAA dataset. This high accuracy was obtained after training for various epochs, ranging from 10 to 40, using the Adam optimizer, categorical cross-entropy loss function, and softmax classifier. The learning rate used was the default of the Adam optimizer, which is 0.001, and the batch size was set to 32. We used diffused images created by setting the smoothness of diffusion (param. alpha) to 15. Table 4 shows the summary of the results obtained with Inception v4 for various epochs.

We initially used diffused images created by setting the smoothness of diffusion (param. alpha) to 25 and experimented with different combinations of the number of inception A–C blocks, with the batch size set to 32. It was observed that the default of four inception A blocks, seven inception B blocks, and three inception C blocks gave better results. The results obtained are summarized in Table 5.

We also conducted experiments by keeping only parts of the architecture. We experimented with using only the inception stem, the inception A blocks, and reduction A block, eliminating blocks B and C. We further tried using only the inception stem followed by blocks A and B, leaving out block C, and also inception stem followed by blocks A and C, leaving out block B. The results in these cases were not as accurate. By using only the inception stem, inception A blocks, and reduction A block, the test accuracies obtained with various numbers of inception A blocks are summarized in Table 6 below.

With the diffused images created by setting the smoothness parameter (param. alpha) to 15, we also performed experiments by using a sample of the training images as a validation dataset. Out of the 3491 training images, we used 3000 images for training and 491 images for validation. When the validation set was incorporated after each epoch during the training process, the model was evaluated on the validation set, which provided an estimate of the skill of the model. Table 7 shows the validation accuracy after each epoch, when the Inception v4 network was trained for 25 epochs, and Figure 7 shows the corresponding plot of epoch versus validation accuracy.

### 4.4. Comparison of the Three Architectures

An overall comparison of the three architectures showed that the Inception v4 had superior performance for face liveness detection compared with ResNet50 and CNN-5. The accuracies of ResNet50 and CNN-5 differed only by a slight amount, and though their accuracies were not as high as the Inception v4 network, they still provided competitive results, showing that they too can be used for face liveness detection of anisotropic diffused captured images. Table 8 shows the summary of the obtained results.

Figure 8 shows the plot of epochs versus accuracy for each architecture, using the results in Table 2, Table 3 and Table 4.

We further observed from our experiments using the three networks that the best optimizer was the Adam optimizer. We tried various values of learning rates for this optimizer to see if increasing or decreasing the learning rate from the default value of 0.001 would improve the test accuracy, but it was observed that the default learning rate gave higher accuracy for ResNet50 and CNN-5. The Inception v4 still had 100% accuracy despite changing the learning rate. Table 9 summarizes the observations made.

We created five sets of diffused images for our experiments, with different values of the parameter param. alpha, which defines the smoothness of the diffused image, as shown in Figure 6. We tested each architecture with each of these sets of diffused images. 

Table 10 summarizes the results obtained with each architecture using each set of diffused images. The entries in parentheses indicate the number of epochs we trained the network to achieve that accuracy. Figure 9 shows the corresponding plot of param. alpha of the diffused images versus the test accuracies obtained with each architecture.

As we have mentioned in our experimental results, with a lower value of the smoothness parameter (param. alpha = 15), Inception v4 had superior performance compared with CNN-5 and ResNet50. Since filters of multiple sizes operate at a level in an inception network, features at different scales can be recognized. Therefore, even with a smoothness parameter of 15, Inception v4 recognized the important features that highlighted the edges and boundary locations to differentiate between a live and fake image. With a higher value of the smoothness parameter (param. alpha = 25), CNN-5 and ResNet50 performed slightly better than the Inception v4 network. However, for still higher values of param. alpha (50, 75, and 100), Inception v4 outperformed CNN-5 and ResNet50. 

In terms of computation speed, ResNet50 was faster than Inception v4 due to the skip connections that allow propagation of the gradients through the shortcut paths by skipping a few convolutional layers. Inception v4 took approximately 20 min to achieve 100% accuracy in 15 epochs, whereas ResNet50 took 20 min to achieve 95.85% accuracy in 20 epochs. CNN-5 took approximately 5 min to achieve the 95.99% accuracy after 30 epochs. This CNN, being only five layers deep and with the added advantage of a reduced number of parameters due to the shared weights concept in CNNs, achieved faster training.

### 4.5. Comparison with State-of-the-Art Methods

We compared the performance of our approach with other state-of-the-art approaches on the NUAA dataset, such as methods that make use of enhanced local binary patterns [1], dynamic local ternary patterns [8], difference of Gaussian filtered images [7], local speed patterns [3], multiscale local binary patterns [9], and a deep CNN proposed in [13]. Compared with these methods, our Inception v4 model achieved 100% accuracy on the test set, making it highly suitable for the classification of a two-dimensional image as being real or fake. 

The Inception v4 network outperformed other state-of-the-art methods in the literature, proving that it is indeed a successful method for use in face recognition applications that use liveness detection as a precursor. The ResNet50 and CNN-5 networks proposed by us did not achieve accuracies as high as the Inception v4 network, but these architectures still gave competitive results when compared to the rest of the methods. Table 11 and Figure 10 confirm these findings.

## 5. Conclusions

In this paper, we developed an optimal solution to the face liveness detection problem. We first applied nonlinear diffusion based on an additive operator splitting scheme and a block-solver tridiagonal matrix algorithm to the captured images. This generated diffused images, with the edge information and surface texture of the real images being more pronounced than fake ones. These diffused images were then fed to a CNN to extract the complex and deep features to classify the images as real or fake. Our implementation with the deep CNN architecture, Inception v4, on the NUAA dataset gave excellent results of 100% accuracy, showing that it can efficiently classify a two-dimensional diffused image as real or fake. Though the CNN-5 and ResNet50 did not give results as good as those of the Inception v4 network, they still returned promising results. A comparative analysis of the three architectures showed that the smoothness of a diffused image is an important factor in determining the liveness of a captured image. We determined that with a low value of this smoothness parameter, the Inception v4 network outperformed the 5-layer CNN and the 50-layer residual network due to its capability of recognizing features at different scales. With a higher value of this smoothness parameter, the 50-layer residual network and the 5-layer CNN performed slightly better than the Inception v4. However, for still higher values of the smoothness parameter, Inception v4 showed better performance. Not only did the Inception v4 outperform the 50-layer residual network and the 5-layer CNN, it also outperformed other state-of-the-art approaches used for face liveness detection. Compared with the Inception v4 network, faster performance was obtained with ResNet50 due to the shortcut paths incorporated in it, and CNN-5 performed still faster because it has fewer layers. Our future work will consist of using recurrent neural networks based on Long Short-Term Memory (LSTM) for face liveness detection on video streams.

## Figures and Tables

**Figure 1 entropy-21-00423-f001:**
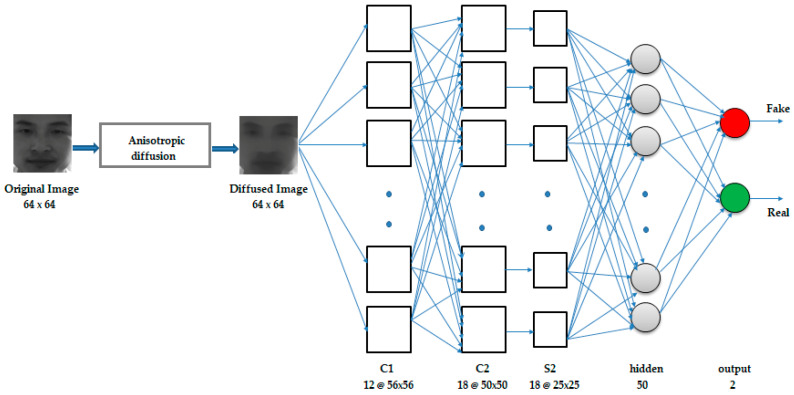
Convolutional neural network (CNN) architecture.

**Figure 2 entropy-21-00423-f002:**
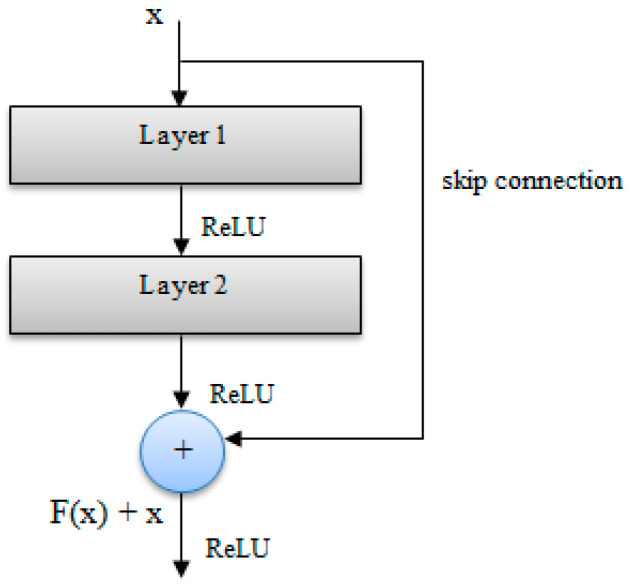
Basic residual block (adapted from [27]).

**Figure 3 entropy-21-00423-f003:**
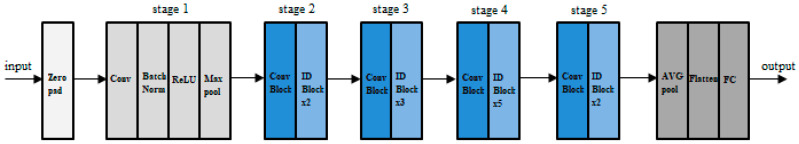
ResNet50 architecture.

**Figure 4 entropy-21-00423-f004:**
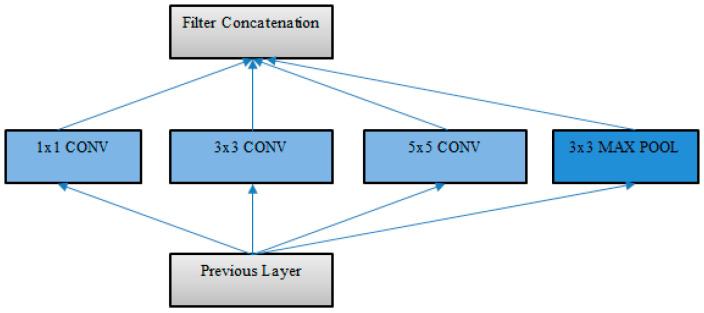
Basic inception module (adapted from [32]).

**Figure 5 entropy-21-00423-f005:**
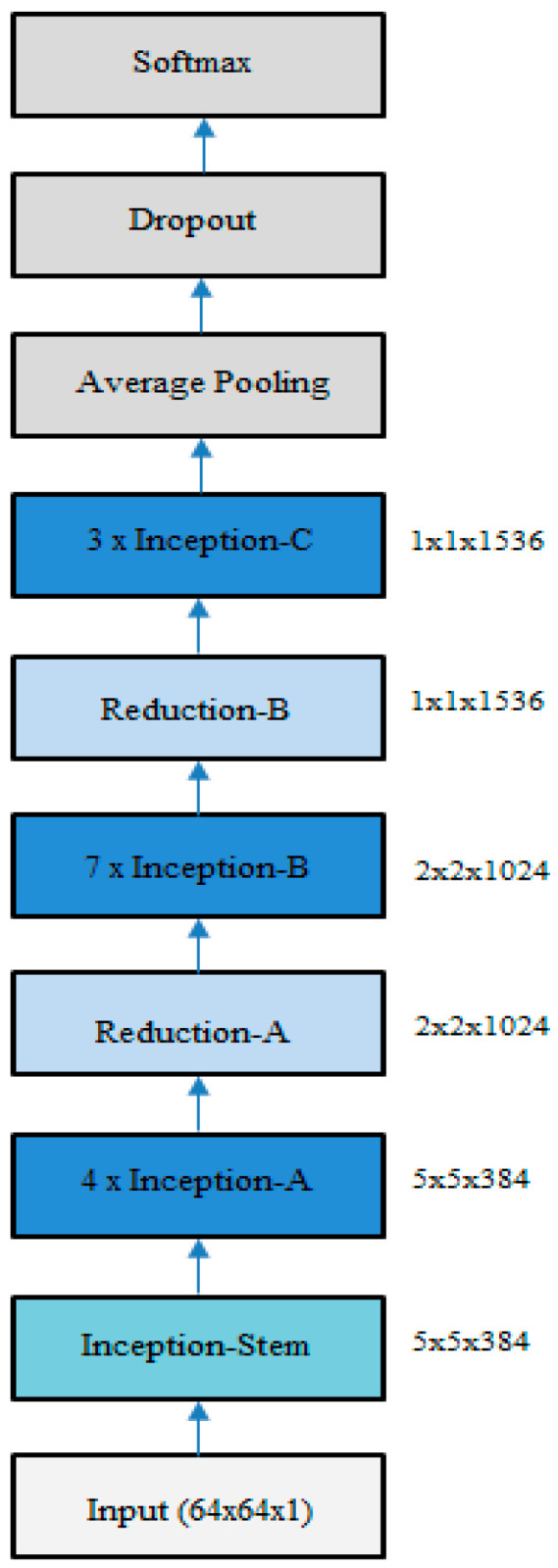
Inception v4 network (adapted from [31]).

**Figure 6 entropy-21-00423-f006:**
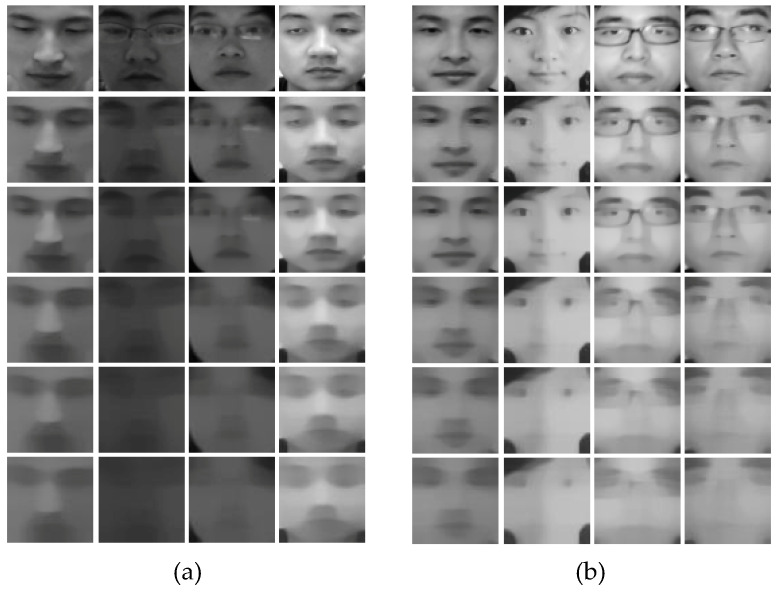
Samples from the NUAA database. (**a**) The top row shows real images in the database, and each row below shows the corresponding diffused images that we created with value of param. alpha set to 15, 25, 50, 75, and 100, respectively. (**b**) The top row shows fake images in the database, and each row below shows the corresponding diffused images created with value of param. alpha set to 15, 25, 50, 75, and 100, respectively.

**Figure 7 entropy-21-00423-f007:**
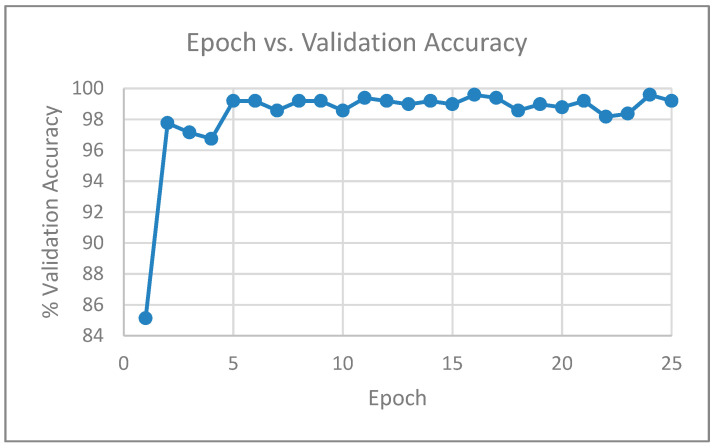
Plot showing epoch vs. validation accuracy for Inception v4.

**Figure 8 entropy-21-00423-f008:**
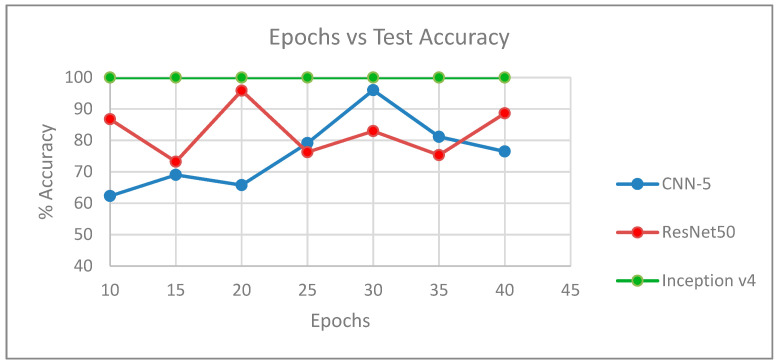
Plot showing epochs vs. accuracy for each architecture.

**Figure 9 entropy-21-00423-f009:**
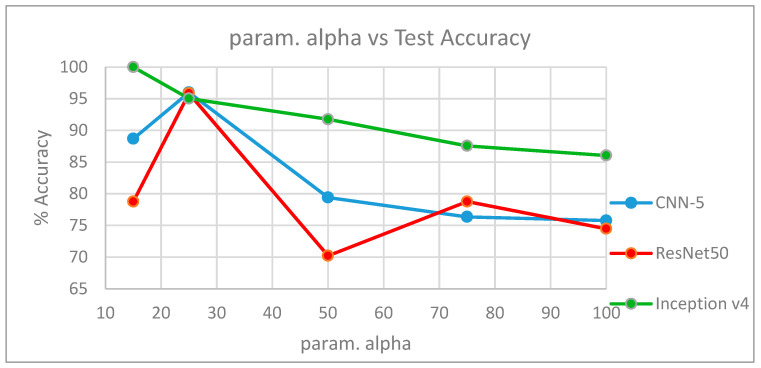
Plot showing param. alpha vs. accuracy for each architecture.

**Figure 10 entropy-21-00423-f010:**
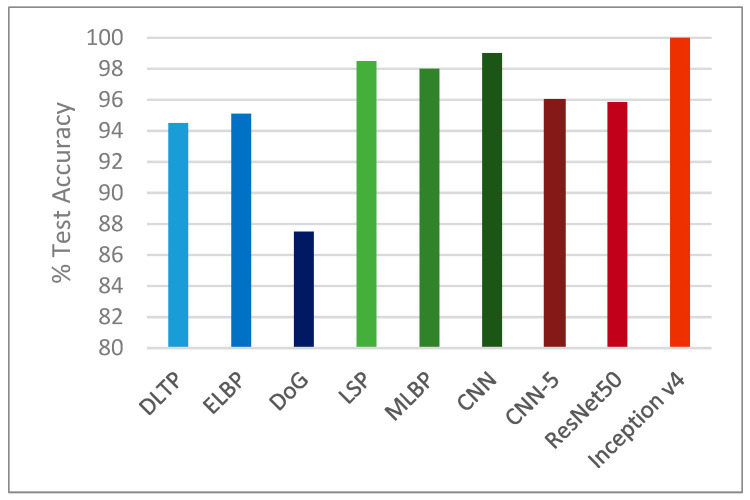
Performance comparison on the NUAA dataset.

**Table 1 entropy-21-00423-t001:** NUAA dataset.

	Training Images	Test Images	Total
Real (Client)	1743	3362	5105
Fake (Imposter)	1748	5761	7509
Total	3491	9123	12,614

**Table 2 entropy-21-00423-t002:** Test results obtained with the CNN-5 architecture.

**Epochs**	10	15	20	25	**30**	35	40
**% Test Accuracy**	62.30	69.04	65.77	79.14	**95.99**	81.15	76.47

**Table 3 entropy-21-00423-t003:** Test results obtained with the ResNet50 architecture.

**Epochs**	10	15	**20**	25	30	35	40
**% Test Accuracy**	86.75	73.21	**95.85**	76.25	82.96	75.32	88.61

**Table 4 entropy-21-00423-t004:** Test results obtained with the Inception v4 architecture.

**Epochs**	**10**	**15**	**20**	**25**	**30**	**35**	**40**
**% Test Accuracy**	**100**	**100**	**100**	**100**	**100**	**100**	**100**

**Table 5 entropy-21-00423-t005:** Evaluation with various numbers of inception A–C blocks (param. alpha = 25).

Number of Blocks Used	Epochs	Test Accuracy
4A, 7B, 3C	10	95.04%
3A, 5B, 2C	10	74.29%
7A, 9B, 5C	10	86.53%

**Table 6 entropy-21-00423-t006:** Evaluation using only inception stem, inception A blocks, and reduction A block (param. alpha = 25).

Number of Inception A Blocks	Epochs	Test Accuracy
4	10	89.07%
5	10	91.17%
6	10	90.91%
7	10	91.59%

**Table 7 entropy-21-00423-t007:** Validation accuracy (%) obtained after each epoch (param. alpha = 15).

Epoch	Accuracy	Epoch	Accuracy	Epoch	Accuracy
1	85.13	11	99.39	21	99.19
2	97.76	12	99.19	22	98.17
3	97.15	13	98.98	23	98.37
4	96.74	14	99.19	24	99.59
5	99.19	15	98.98	25	99.19
6	99.19	16	99.59		
7	98.57	17	99.39		
8	99.19	18	98.57		
9	99.19	19	98.98		
10	98.57	20	98.78		

**Table 8 entropy-21-00423-t008:** Comparison of the three architectures.

Architecture	Epochs	Optimizer	Loss Function	Test Accuracy
CNN-5	30	Adam	mean squared error	95.99%
ResNet50	20	Adam	categorical cross-entropy	95.85%
Inception v4	10	Adam	categorical cross-entropy	100%

**Table 9 entropy-21-00423-t009:** Accuracies obtained for various values of learning rates.

Architecture	Learning Rate	Epochs	Accuracy
CNN-5	**0.001**	**30**	**95.99%**
CNN-5	0.002	30	88.97%
CNN-5	0.005	30	63.14%
CNN-5	0.0008	30	72.83%
ResNet50	**0.001**	**20**	**95.85%**
ResNet50	0.002	20	62.74%
ResNet50	0.005	20	64.80%
ResNet50	0.0008	20	66.76%
Inception v4	**0.001**	**10**	**100%**
Inception v4	0.002	10	**100%**
Inception v4	0.005	10	**100%**
Inception v4	0.0008	10	**100%**

**Table 10 entropy-21-00423-t010:** Accuracies obtained with each set of diffused images (the epochs are shown in parentheses).

Param. alpha	CNN-5	ResNet50	Inception v4
15	88.69% (25)	78.76% (40)	**100% (10)**
25	**95.99% (30)**	**95.85% (20)**	95.04% (10)
50	79.41% (20)	70.21% (30)	91.76% (5)
75	76.35% (20)	78.77% (30)	87.56% (1)
100	75.77% (30)	74.47% (30)	86.04% (5)

**Table 11 entropy-21-00423-t011:** Comparison of our proposed method with other state-of-the-art methods on the NUAA dataset.

Method	Test Accuracy
ELBP [1]	95.1%
DLTP [8]	94.5%
DoG [7]	87.5%
LSP [3]	98.5%
MLBP [9]	98%
CNN [13]	99%
CNN-5 (proposed method)	95.99%
ResNet50 (proposed method)	95.85%
Inception v4 (proposed method)	**100%**

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
