# Peer review of "Optimizing Deep CNN Architectures for Face Liveness Detection"

_entropy, 2019, doi:10.3390/e21040423_

Round 1

Reviewer 1 Report

The aim of this paper is interesting and useful for readers thanks to the comparison's work. 

For a better reading, the authors must re-read the paper to better introduce the name of the parameters and better to define for example param.alpha, Theta and x ...

In paragraph 3.2 give the references of the mentioned fields.

The introduction of the three architectures is not well detailed.

Why do you take the first 9 subjects for the learning phase and not a random one?

Why are not you interested in LSTM algorithm? It will be interesting for video streams.

Finally, websites should be avoided for references.

Author Response

Point 1: The aim of this paper is interesting and useful for readers thanks to the comparison's work.

Response1: We thank the reviewer for reviewing the paper, and the positive assessment.

Point 2: For a better reading, the authors must re-read the paper to better introduce the name of the parameters and better to define for example param.alpha, Theta and x …

Response 2: We have made updates in the revised manuscript by providing a better definition of param. alpha in the third paragraph of Introduction, defining Theta in the equations in 3.1, and defining x in the second paragraph of 3.2.1.

Point 3: In paragraph 3.2 give the references of the mentioned fields.

Response 3: We are sorry for missing the references. We have updated it in the revised manuscript.

Point4: The introduction of the three architectures is not well detailed.

Response 4: We apologize for not providing sufficient detail in the introduction of the architectures. We have updated it in the revised manuscript.

Point 5: Why do you take the first 9 subjects for the learning phase and not a random one?

Response 5: The NUAA dataset consists of 3491 training images and 9123 test images. The training set images consists of subjects 1-9, and the test set consists of subjects 1-15. Since the training set already consists of subjects 1-9, we used that in the learning phase.

Point 6: Why are not you interested in LSTM algorithm? It will be interesting for video streams.

Response 6: We have mentioned this in the Conclusion of our paper.

Point 7: Finally, websites should be avoided for references.

Response 7: We changed it in the revised manuscript, except for references [30] and [31]. These are arxiv publications, for which websites are generally accepted in technical papers.

Reviewer 2 Report

This paper conducted a study to differentiate real and fake face images with nonlinearly diffused images and three deep CNN architectures, including CNN-5, ResNet50 and Inception v4. The NUAA dataset consisting of 15 subjects and 12614 images was tested. Different smoothness and CNN parameters were tried. Inception v4 combined with smoothness factor 15 resulted in a 100% classification rate.

1. The abstract is not clear and needs to be improved.

2. Why the images need to be diffused for CNN? Please clarify if it is for denoising. What if the original images are used for CNN?

3. More details are needed for the proposed methods.

Author Response

Point 1: This paper conducted a study to differentiate real and fake face images with nonlinearly diffused images and three deep CNN architectures, including CNN-5, ResNet50 and Inception v4. The NUAA dataset consisting of 15 subjects and 12614 images was tested. Different smoothness and CNN parameters were tried. Inception v4 combined with smoothness factor 15 resulted in a 100% classification rate.

Response 1: We thank the reviewer for taking time to review our paper, and for the suggestions.

Point2: The abstract is not clear and needs to be improved.

Response 2: We have improved it in the revised manuscript.

Point 3: Why the images need to be diffused for CNN? Please clarify if it is for denoising. What if the original images are used for CNN?

Response 3: When diffusion is used, the edge information and surface texture of real images will be more pronounced than that of fake images. The diffusion process denoises the image by preserving the edges. The edges of a flat image will fade out, whereas those of a real image will remain clear. This helps in classification of the image as live or fake. Without diffusion, the edges will not be highlighted and the boundaries will not be enhanced. Therefore, when an original image is fed to the architecture, classification accuracy will be very low as it will be difficult to distinguish between a real and fake image.

Point4: More details are needed for the proposed methods.

Response 4: We apologize for not including sufficient detail in the proposed method. We have added more information in the revised manuscript.